# Epigenetic Memories in Hematopoietic Stem and Progenitor Cells

**DOI:** 10.3390/cells11142187

**Published:** 2022-07-13

**Authors:** Kazumasa Aoyama, Naoki Itokawa, Motohiko Oshima, Atsushi Iwama

**Affiliations:** Division of Stem Cell and Molecular Medicine, Center for Stem Cell Biology and Regenerative Medicine, The Institute of Medical Science, The University of Tokyo, 4-6-1, Shirokanedai, Minato-ku, Tokyo 108-8639, Japan; itokawanaoki@g.ecc.u-tokyo.ac.jp (N.I.); moshima@ims.u-tokyo.ac.jp (M.O.)

**Keywords:** next-generation sequencing, epigenetic memory, epigenome, chromatin accessibility, hematopoietic stem cells, hematopoietic progenitor cells, innate immune cells

## Abstract

The recent development of next-generation sequencing (NGS) technologies has contributed to research into various biological processes. These novel NGS technologies have revealed the involvement of epigenetic memories in trained immunity, which are responses to transient stimulation and result in better responses to secondary challenges. Not only innate system cells, such as macrophages, monocytes, and natural killer cells, but also bone marrow hematopoietic stem cells (HSCs) have been found to gain memories upon transient stimulation, leading to the enhancement of responses to secondary challenges. Various stimuli, including microbial infection, can induce the epigenetic reprogramming of innate immune cells and HSCs, which can result in an augmented response to secondary stimulation. In this review, we introduce novel NGS technologies and their application to unraveling epigenetic memories that are key in trained immunity and summarize the recent findings in trained immunity. We also discuss our most recent finding regarding epigenetic memory in aged HSCs, which may be associated with the exposure of HSCs to aging-related stresses.

## 1. Introduction

The recent development of next-generation sequencing (NGS) technologies has enabled genome-wide analysis of various biological processes at single-cell resolution. Single-cell RNA sequencing (scRNA-seq) provides gene expression profiles at single-cell resolution. Assay for transposase-accessible chromatin sequencing (ATAC-seq) identifies the open chromatin regions associated with epigenetic regulation and can be applied to single-cell analysis. Chromosome conformation capture (Hi-C) techniques map the 3D organization of entire genomes. This technology has also been optimized for single-cell analysis. These novel NGS technologies are powerful tools used to decipher unsolved biological phenomena [1,2,3]. Trained immunity is one of the biological phenomena recently unraveled by NGS-based epigenetic analysis of HSCs and innate immune cells, and involves their epigenetic reprogramming upon primary stimulation, which can result in an augmented response to secondary stimulation. In this review, we summarize the recent progress in the research of epigenetic memories inscribed in hematopoietic stem and progenitor cells [4,5].

## 2. New NGS Technology in Hematopoiesis Research

### 2.1. RNA Sequencing

RNA sequencing, which is usually performed on more than thousands of cells, has been widely utilized to analyze gene expression patterns over the past decades. In bulk RNA-seq analysis, only average gene expression levels of collected cells can be obtained (Figure 1A). Since gene expression patterns are heterogeneous, even in similar types of cells [6,7,8], and these differences in the gene expression landscape determine the composition of cell types and decide cell fate [9,10], there is an increasing demand for analysis on the gene expression levels of individual cells.

scRNA-seq, which was developed in 2009 [11], provides us with genome-wide gene expression profiles at single-cell resolution [12,13,14,15]. The ability of the scRNA-seq technique to analyze differences in gene expression landscapes between individual cells potentially uncovers rare cell populations that cannot be detected by bulk RNA-seq analysis (Figure 1B). For example, the single-cell technique identified a rare population of cells displaying cancer drug resistance with high expression levels of resistance markers [16]. Although single-cell RNA-seq has limitations, including the production of a high level of technical noise due to the low amount of starting materials compared with bulk RNA-seq [17], recent advances in experimental techniques and bioinformatics have provided various findings, such as transcriptional bimodality in immune cells [18], dynamics and regulators of cell fate decisions [19], lineage and X chromosome dynamics in preimplantation embryos [20], T cell fate and clonality inference [21], and heterogeneity of HSC behavior [22]. Additionally, scRNA-seq combined with single-cell proteomics has been applied to study the systemic immune responses to SARS-CoV-2 infection [23].

### 2.2. ATAC Sequencing

Recently, studies on gene regulation have focused on epigenetics because the advances in sequencing technologies have resulted in the development of various assays for epigenome analysis. Such epigenome assays include chromatin accessibility assays (assay for transposase-accessible chromatin sequencing (ATAC-seq)) [24,25,26,27,28], DNase I hypersensitive sites sequencing (DNase-seq) [29,30,31], formaldehyde-assisted isolation of regulatory elements sequencing (FAIRE-seq) [32,33], transcription factor-binding and histone modification assays (chromatin immuno-precipitation sequencing (ChIP-seq) [34,35,36,37]), and nucleosome positioning and occupancy assays (micrococcal nuclease sequencing (MNase-seq)) [38,39].

Since first described in 2013, the number of ATAC-seq datasets and publications is markedly increasing compared with that of other chromatin accessibility assays [28]. In ATAC-seq, hyperactive Tn5 transposase is used to access open chromatin regions and to simultaneously cut and ligate adaptors (Figure 1C). Genome-wide mapping of ligated adaptors by high-throughput sequencing enables us to identify the open accessible chromatin regions [25]. ATAC-seq analysis, such as RNA-seq, can be carried out at the single-cell level (single-cell ATAC-seq, scATAC-seq) [40,41,42,43], as well as on pooled cells (bulk ATAC-seq). ATAC-seq has provided biologically meaningful insights into various situations, including in embryonic development and disease research [27,44,45], as well as hematopoietic differentiation [1,46].

### 2.3. Hi-C Technique

The chromosome conformation capture (Hi-C) technique measures pairwise contact frequencies between all loci in the genome. Hi-C data provide higher-order genome organization, including the formation of chromatin loops and topologically associating domains, and have been used to build three-dimensional chromatin models (Figure 1D) [47,48,49,50]. However, it should be noted that Hi-C data represent an average of target cells, and that nuclear genome organization and chromosome structure are not uniform, even among cells of a homogeneous population.

To overcome this limitation, Hi-C technology has been optimized for single-cell analysis [51,52]. Although the interactome maps produced by single-cell Hi-C are sparse, the single-cell Hi-C technique was successfully applied to understand cellular variability in nuclear genome organization and chromosome structure during cell cycle and in early embryogenesis [2,53].

**Figure 1 cells-11-02187-f001:**
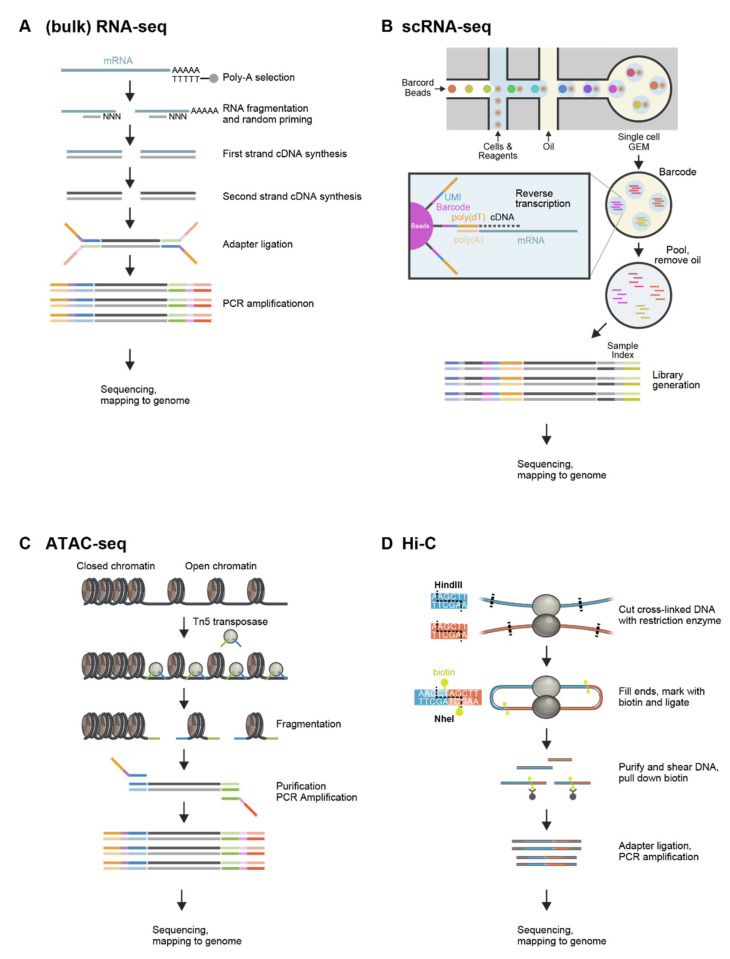
Overview of NGS analyses. (**A**) In (bulk) RNA-seq, poly-A-selected total RNA undergoes cDNA synthesis and adapter ligation, followed by PCR amplification. The resulting library is subjected to sequencing and mapping to the genome. (**B**) In scRNA-seq with 10× Genomics Next GEM Technology, barcoded Gel Beads mixed with the cells, reagents, and partitioning oil within chromium instrument form GEMs (Gel Bead-in-emulsion) that contain a single cell. cDNAs are synthesized and barcoded in each GEM, then pooled to generate library for sequencing. (**C**) In ATAC-seq with a Tn5 transposase-based method, Tn5 transposases bind open chromatin regions and cut and ligate adaptors. After purification, PCR amplification, and sequencing, ligated adaptors are mapped to the genome. (**D**) In Hi-C analysis, formaldehyde generates cross-links between spatially adjacent DNA. After digestion with a restriction enzyme, the sticky ends are filled, marked with biotin, and ligated. The resulting DNA is purified, sheared, followed by isolation for biotin-marked DNA. Isolated DNA is subjected to sequencing and mapping to the genome [54].

## 3. Trained Immunity and Epigenetic Memory

Classically, immunological memory is known to be carried out by T and B lymphocytes, which are responsible for adaptive immune systems. However, recently, innate system cells, such as macrophages, monocytes, and natural killer cells, and their bone marrow (BM) progenitors have been found to gain memories upon transient stimulation, leading to the enhancement of response to secondary challenges. This phenomenon has been termed “trained immunity” [4,5,55]. Various stimuli, including microbial infections, can activate innate immune cells and induce a primary response. At the same time, the first stimuli can induce the reprogramming of immune cells, which can result in an augmented response to secondary stimulation (Figure 2). Although trained immunity was first discovered in circulating innate immune cells, such as monocytes and NK cells, they have a short lifespan. Recent work has resolved that HSCs can directly respond to acute and chronic infections. Innate immune cells derived from trained HSCs migrate to peripheral organs, where they undergo differentiation/maturation and exert enhanced effector functions against pathogens. Therefore, trained immunity can occur in BM progenitor cells (central trained immunity) as well as in blood innate immune cells (peripheral trained immunity) [4,5]. In contrast to the adaptive immune system, trained immunity utilizes the epigenetic reprogramming of transcriptional pathways, as well as metabolic reprogramming, as the memory of the first stimulation [5,56,57].

The first stimuli can induce the activation of immune cells, which results in the activation of gene transcription accompanied by the rapid acquisition of active histone modifications at promoters and enhancers. After removal of the stimulus, some of these histone modifications persist, which leads to faster and augmented activation of gene transcription upon secondary stimulation. The histone modifications involved in trained immunity are histone H3 lysine 4 tri-methylation (H3K4me3), which marks active promoters; histone H3 lysine 4 mono-methylation (H3K4me1), which marks distal enhancers; and histone H3 lysine 27 acetylation (H3K27ac), which marks both active enhancers and promoter regions [4,5]. Chromatin accessibility is another important factor involved in epigenetic memory [29,58] (Figure 3).

The activity of enhancers is associated with marks with histone modifications, as well as with the recruitment of transcription factors [59]. Although several histone modifications have been shown to have critical roles for enhancer activity, usually H3K4me1, H3K27ac, and H3K27me3 are the major histone modifications observed in enhancers and used to indicate the activity of enhancers. The features of active enhancers were originally identified by the Encyclopedia of DNA Elements (ENCODE) project [60]. H3K4me1, mediated by MLL3/MLL4, SET7/SET9 (writer), KDM1A, and KDM5C (eraser), is the basic mark of enhancers. H3K27ac, mediated by CBP/p300 (writer), is used as the mark of active enhancers [61]. On the other hand, H3K27me3, mediated by EZH1/EZH2 (writer) and KDM6A/KDM6B (eraser), is the mark of silenced enhancers. The chromatin regions marked with “H3K4me1 and H3K27me3”, “only H3K4me1”, and “H3K4me1 and H3K27ac” are defined as “poised enhancer”, “primed enhancer”, and “active enhancer”, respectively. In addition, the enhancers that retain relaxed chromatin but do not have any of these histone modifications are termed “inactive enhancers” [59,62,63] (Figure 3).

Histone acetylation at active enhancers promotes the recruitment of transcriptional machinery to chromatin by serving as a scaffold. Acetylated lysine is recognized by bromodomains, which are present in various transcriptional coactivators, including histone acetyl transferases themselves (e.g., CBP/p300), ATP-dependent chromatin remodelers (e.g., SWI/SNF complexes members), general transcription factors (e.g., TAF1), and the bromodomain and external domain (BET) family of proteins involved in productive RNA polymerase II elongation (e.g., BRD2, BRD4) [64]. Bromodomain-containing proteins are recruited to chromatin through their association with acetylated lysine, and then function as scaffold proteins to recruit additional transcriptional machinery. For example, BET proteins recruit the positive elongation factor b (P-TEFb), which leads to the phosphorylation of RNA polymerase II and the stimulation of transcription from the promoter [65,66]. The recruitment of proteins and complexes mediated by acetylated lysine at enhancers and promoters might result in a feed-forward loop that stabilizes transcription [67].

**Figure 3 cells-11-02187-f003:**
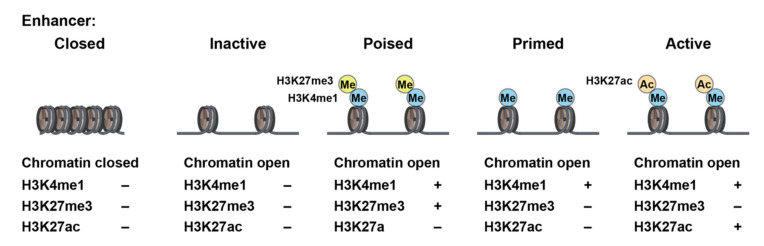
Histone marks and enhancer activity. The histone modifications, H3K4me1, H3K27me3, and H3K27ac, and chromatin accessibility are associated with enhancer activity. Inactive, poised, primed, and active enhancers are defined as H3K4me1+/H3K27me3+/H3K27ac−, H3K4me1+/H3K27me3−/H3K27ac−, and H3K4me1+/H3K27me3−/H3K27ac+ open chromatin regions, respectively [59,62,63].

## 4. Epigenetic Memory in Hematopoietic Stem and Progenitor Cells

Novel NGS technologies, such as scRNA-seq, bulk ATAC-seq, and scATAC-seq, have been utilized in not only immune cells, but also hematopoietic stem and progenitor cells (HSPCs), revealing the roles of epigenetic remodeling in epigenetic memories induced by various stimuli (Figure 4).

### 4.1. Epigenetic Memory in Hematopoietic Stem and Progenitor Cells

Divangahi and colleagues showed that the access of Bacillus Calmette–Guérin (BCG) to the bone marrow (BM) educates HSPCs for the establishment of trained immunity [69] (Figure 4A). Previously, they and other groups demonstrated the essential roles of macrophages in protection against infection with *Mycobacterium tuberculosis* (*Mtb*) [58,74,75]. Vaccination with BCG expanded HSPCs and enhanced myelopoiesis in mice. scRNA-seq analysis of lineage^−^ Sca-1^+^ c-Kit^+^ (LSK) HSPCs revealed that the transcription factors involved in the response to type II interferon (IFN-II) IFNγ (e.g., *Stat1* and *Irf1*) were up-regulated in the majority of LSK cells. This result was supported by the finding that BM HSPCs were not expanded in response to BCG vaccination in IFNγ receptor–deficient mice. ChIP-seq for histone modifications (H3K27ac and H3K4me1) demonstrated that BCG vaccination–trained HSCs generate macrophages epigenetically primed to initiate a more protective response against *Mtb* infection, priming the key enhancers of the genes involved in the response to the *Mtb* infection poised/active state. Collectively, these results suggest that the transcriptional reprogramming of HSCs induced by IFNγ is involved in BCG vaccination–induced trained immunity. BCG-induced trained immunity via HSPCs has also been confirmed in humans [69,76].

*Mtb* can disseminate to the BM in patients with tuberculosis [77]. However, the effects of *Mtb* on HSCs and trained immunity in the BM remained unknown. The same group demonstrated that *Mtb*, in contrast with BCG, induces the suppression of myelopoiesis, resulting in impaired trained macrophage immunity through the altered transcriptional landscape of HSCs [70] (Figure 4B). scRNA-seq analysis of LSK HSPCs showed that IFN-I and II pathways were activated by *Mtb* infection. As described above, IFN-II signaling is critical for HSC expansion and myelopoiesis [69,78,79], whereas IFN-I signaling causes a loss of stem cell numbers and function [80,81]. An analysis of *Ifnar1^−/^*^−^ mice demonstrated the dependency on IFN-I signaling for the suppression of myelopoiesis and immune training during infection with *Mtb*. Collectively, these results indicate that transcriptional reprogramming of HSCs by BCG and *Mtb* leads to protective and failed trained immunity, respectively. Notably, these effects last for at least 1 year [70].

### 4.2. β-Glucan–Induced Trained Immunity in Neutrophils and BM GMPs

Similar to BCG, fungal-derived polysaccharide β-glucan promotes the sustained enhanced response of myeloid cells to secondary infections or inflammatory challenges. These agonists for trained immunity also have anti-tumor activities. The efficacy of β-glucan in tumor immune therapy has been reported [82,83,84,85,86]. However, how trained immunity agonists impact anti-tumor responses and whether trained immune memory is involved in the tumor-suppressive functions of the agents, such as β-glucan, are not well understood.

Kalafati and colleagues demonstrated that the pre-treatment of mice with a single dose of β-glucan significantly delayed the progression of tumors [72] (Figure 4D). The adoptive transfer of neutrophils from β-glucan–trained mice to naive recipient mice suppressed tumor growth, suggesting that the anti-tumor effect of immune training induced by β-glucan involves, at least in part, the training of granulopoiesis. Moreover, the anti-tumor effect of trained granulopoiesis induced by β-glucan was transmissible by BM transplantation to naive recipient mice.

Here, scATAC-seq of neutrophiles and BM GMPs revealed that the enhanced chromatin accessibility of the regions that were linked to the IFN-I signaling-related genes, and the motif of IRF1 transcription factor, was enriched in the differentially open accessible regions (open DARs) induced by β-glucan in GMPs. Taken together with the finding that *Ifnar1*-deficient mice treated with β-glucan failed to train granulopoiesis to repress tumor growth, these results suggest that β-glucan promotes anti-tumor activity through epigenetic changes in granulopoiesis that are dependent on IFN-I signaling.

### 4.3. LPS-Induced Trained Immunity in HSCs

De Laval and colleagues showed that lipopolysaccharide (LPS)-induced acute immune stimulation establishes epigenetic memory in HSCs, which leads to better response to secondary stimulation in a manner dependent on myeloid transcription factor C/EBPβ [71] (Figure 4C). The ability of HSCs to respond to LPS exposure, which is a mimic of bacteria, has been reported by other groups [87,88]. To examine whether exposure of HSCs to LPS has an impact on the secondary challenge that occurs later, HSCs were harvested and transplanted from CD45.1 mice treated with LPS to lethally irradiated CD45.2 recipient mice. Upon infection with *P aeruginosa*, the recipient mice with LPS-pre-exposed HSCs exhibited significant decreased mortality compared with those with control HSCs. Even HSCs from mice 13 weeks after exposure to LPS exhibited a similar protective effect on recipient mice upon infection challenge. Of note, an ATAC-seq analysis of HSPCs revealed that changes in chromatin accessibility were sustained at least 4 weeks after exposure to LPS. The open chromatin regions induced by LPS treatment were enriched for C/EBPβ targets, and C/EBPβ deletion erased the long-term inscription of epigenetic marks induced by LPS. Thus, LPS challenge can induce C/EBPβ-dependent chromatin accessibility, resulting in trained immunity in HSCs during secondary infection.

### 4.4. Epigenetic Memory in Aged HSPCs

As described above, HSPCs are highly responsive to various stresses, such as infection, inflammation, and myeloablation [89,90], which facilitate myelopoiesis by activating HSPCs to undergo precocious myeloid differentiation and transiently amplifying myeloid progenitors that rapidly differentiate into mature myeloid cells [91,92]. This reprogramming of HSPCs, termed “emergency myelopoiesis”, immediately serves to replenish mature myeloid cells to control infection and regeneration [93]. Emergency myelopoiesis is driven via the activation of key myeloid transcriptional networks at the HSPC and myeloid progenitor cell levels [94]. Moreover, HSCs are assumed to be exposed to additional stresses in the aging-associated proinflammatory milieu and dysfunctional aged BM niche. These stresses may be inscribed in HSCs with aging.

Various epigenome alterations, which are dependent on both intrinsic and extrinsic factors, are associated with aging [95,96]. Analyses of DNA methylation and histone modifications on aged HSCs have been reported [97,98,99]. While accumulating evidence for the significance of chromatin accessibility in aged HSCs as well as hematological malignancies has emerged [1,100,101], the alterations in chromatin accessibility associated with aging in HSCs have not been examined. To understand the alterations in the epigenetic status of HSCs with aging, we performed an ATAC-seq analysis of aged HSCs, multipotent progenitors (MPPs), and myeloid progenitors, uncovering epigenetic traits inscribed in chromatin accessibility in aged HSCs [102]. Of interest, alterations in chromatin accessibility were predominantly induced in HSCs within the HSPC fraction with aging, which gradually resolved as the HSCs differentiated, particularly in MPP2, leaving moderate differences in the MPP fraction and downstream progenitors. Open DARs in aged HSCs were largely located at enhancers and showed enrichment of binding motifs of the ATF family (c-Jun-CRE, Atf1, Atf2), STAT family (Stat1, Stat3, Stat5), and CNC family (Nrf2, NF-E2, Bach1, Bach2) transcription factors (Figure 5A). These transcription factors are activated in response to external stimuli [103,104,105]. The JAK-STAT pathway is activated by cytokine signals such as the cytokines that support HSCs, as well as by the inflammatory cytokines that affect HSC function, such as IL-6 and interferons. IL-1β, an inflammatory cytokine, also indirectly activates the JAK-STAT pathway. Atf2, which forms a homodimer as well as a heterodimer with c-Jun, is activated by SAPKs (p38 and JNK) in response to various stresses, such as DNA damage, environmental stresses, and inflammatory cytokines [105]. Nrf2 is a master regulator of the antioxidant response and is essential for maintaining HSCs and their regenerative response to stresses [103]. The enrichment of these motifs at open DARs implies the ongoing and/or past exposure of HSCs to such stresses.

Profiling of H3K4me1, H3K27ac, H3K4me3, and H3K27me3 in young and aged HSCs revealed that most open DARs in aged HSCs were subdivided into active (H3K4me1+/H3K27ac+), primed (H3K4me1+/H3K27ac−), and inactive (H3K4me1−/H3K27ac−) enhancers. These results indicate that open DARs in aged HSCs comprise active enhancers induced by ongoing stresses, and primed and inactive enhancers that can be readily activated by re-challenge to their upstream stresses. In good agreement with these findings, genes linked to open DARs showed significantly higher levels of basal expression and significantly higher expression peaks after cytokine stimulation in aged HSCs compared with young HSCs, suggesting that open DARs contribute to augmented transcriptional responses under stress conditions. These results indicate that a significant portion of open DARs in aged HSCs represent epigenetic memory inscribed by exposure to external stresses in HSCs to augment their responses to secondary external stimuli (Figure 5B). These results clearly indicate that DAR-linked augmented stress responses define the unique behaviors of aged HSCs [102]. Sustained external signals are detrimental for HSCs and the maintenance of hematopoietic homeostasis, resulting in HSC depletion or transformation. It would be intriguing to examine whether dysregulated transcription mediated by DARs in aged HSCs is involved in stress-induced HSC depletion or transformation.

### 4.5. Epigenetic Memory in Non-Immune Cells

Recently, an increasing number of studies have revealed the role of epigenetic memories not only in immune cells, but also in non-immune cells, including tissue stem cells [57,106,107,108,109]. For instance, Fuchs and colleagues revealed an epigenetic mechanism underlying inflammatory memory in epidermal stem cells (EpSCs) using ATAC-seq [73] (Figure 4E). They identified >1000 inflammatory memory domains in EpSCs from mice with imiquimod (IMQ)-induced skin inflammation, which were defined as chromatins that became accessible during inflammation and retained accessibility even after inflammation [110]. Inflammatory memory on the domains was established by general stress-responsive transcription factor FOS, its partner JUN, and stimulus-specific transcription factor STAT3 during inflammation. After inflammation, JUN remained on the domain and maintained inflammatory memory with other homeostatic transcription factors, which contributed to the rapid recruitment of FOS and gene activation upon secondary challenge.

The same group also demonstrated that hair follicle stem cells (HFSCs) expand potency and alter tissue fitness by accumulating diverse epigenetic memories during wound healing [111]. Monitoring the response to wounds, they found that HFSCs leave their niche, migrate to repair damaged epidermis, and take up residence there as epidermal stem cells (EpdSCs). The fate-changed EpdSCs were indistinguishable from native unwounded EpdSCs in homeostatic function and transcriptome. However, ATAC-seq revealed that, at each step, they temporally change chromatin states and accumulate long-lasting epigenetic memories. Long-lasting epigenetic memories of quiescent HFSCs (memory of niche origin), wound response (memory of wound), and adaptation to the new epidermal niche (compensatory adaptation) enable these immigrant stem cells to perform their new tasks with heightened responses to injury, inflammation, and hair regeneration.

## 5. Conclusions

Novel epigenetic technologies, including single-cell approaches, have unveiled cellular signaling cascades in hematopoietic cells for trained immunity that involve epigenetic alterations such as chromatin accessibility and histone modifications. As is the case with aberrant DNA methylation events in aged cells and cancers [44,45], many epigenomic alterations may be merely passenger events with little biological meaning. Therefore, it is challenging to distinguish the epigenomic alterations that are critical for biology from those that are passengers. Further development of the technologies for epigenetic analysis is expected to provide a better understanding of the epigenetics of trained immunity in hematopoietic cells as well as epigenetic memories in general.

## Figures and Tables

**Figure 2 cells-11-02187-f002:**
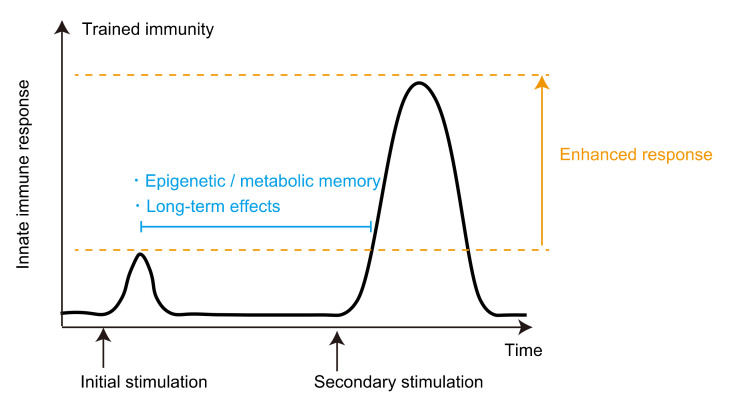
Trained immunity. Initial stimulations can trigger innate immune responses and establish long-term memories via epigenetic and metabolic alterations. Secondary stimulations induce enhanced innate immune responses, leading to better resistance to diseases [4,5].

**Figure 4 cells-11-02187-f004:**
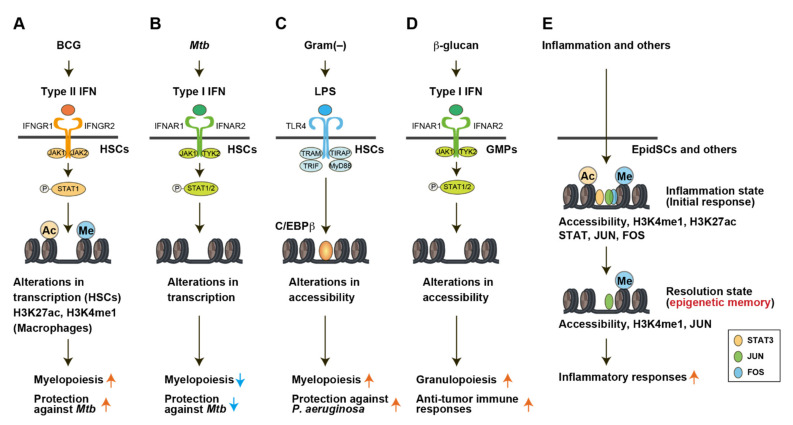
Signaling pathways that induce trained immunity through epigenetic reorganization. Novel technologies, such as scRNA-seq and ATAC-seq, have unveiled signaling pathways for trained immunity through epigenetic reorganization [68]. (**A**) Bacillus Calmette–Guérin (BCG)-induced trained immunity involves type II IFN-mediated alterations in transcription, H3K27ac, and H3K4me1 [69]. (**B**) *Mycobacterium tuberculosis* (*Mtb*)-induced trained immunity involves type I IFN-mediated alterations in transcription [70]. (**C**) Lipopolysaccharide (LPS)-induced trained immunity involves TRL4-mediated alterations in chromatin accessibility that is dependent on C/EBPβ [71]. (**D**) β-glucan-induced trained immunity involves type I IFN-mediated alterations in chromatin accessibility [72]. (**E**) Inflammation-induced trained immunity involves alterations in chromatin accessibility, H3K27ac, and H3K4me1 in epidermal stem cells (EpSCs) [73].

**Figure 5 cells-11-02187-f005:**
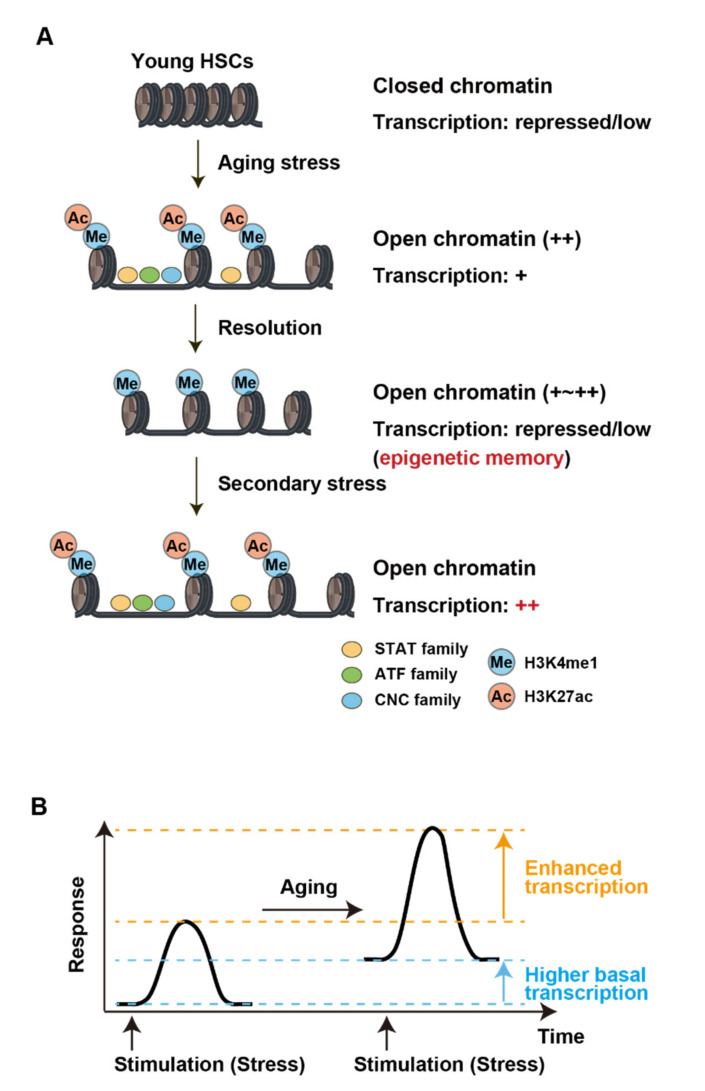
Epigenetic memory established by aging. (**A**) Aging stress-induced epigenetic memory inscribed in chromatin accessibility and histone modifications (H3K4me1 and H3K27ac). The transcription factors, the binding sites of which were enriched in DARs, are depicted. (**B**) Differences in transcriptional responses of open DAR-linked genes between young and aged HSCs [102].

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
