# Peer review of "Epigenetic Memories in Hematopoietic Stem and Progenitor Cells"

_cells, 2022, doi:10.3390/cells11142187_

Round 1
Reviewer 1 Report
In this review the authors clearly describe epigenetic alteration in immune system and focus on hematopoietic stem cell and progenitors alterations. I found this review well written and its content interesting in this field. The figures are appropriate. I just have two suggestions:
Better describe and create a scheme showing hematopoietic signaling triggered by cytokines (jak/stat, PI3k, PLCg) and epigenetic alterations
The authors also would comment about epigenetic alteration by Sars-2, item 4,5
for example
Severe COVID-19 Is Marked by a Dysregulated Myeloid Cell Compartment. Cell. 2020 182(6):1419-1440.e23. doi: 10.1016/j.cell.2020.08.001.
Author Response
In this review the authors clearly describe epigenetic alteration in immune system and focus on hematopoietic stem cell and progenitors alterations. I found this review well written and its content interesting in this field. The figures are appropriate. I just have two suggestions:
Better describe and create a scheme showing hematopoietic signaling triggered by cytokines (jak/stat, PI3k, PLCg) and epigenetic alterations
Re: Thank you for your suggestion. We indicated signaling molecules to which we referred in the manuscript in Figure 3 (currently Figure 4).
The authors also would comment about epigenetic alteration by Sars-2, item 4,5 for example
Severe COVID-19 Is Marked by a Dysregulated Myeloid Cell Compartment. Cell. 2020 182(6):1419-1440.e23. doi: 10.1016/j.cell.2020.08.001.
Re: We mentioned the paper regarding epigenetic alterations by COVID-19.
Reviewer 2 Report
In this rewiev article the authors summarized progress in the research of epigenetic memories in hematopoietic stem cells and progenitor cells.
The layout of the paper is typical for review articles. The choice of content does not raise objections. Citations are adequate to the content and clear. The 107 references are mostly from recent years. Due to some phrases it requires re-reading and correction by an English native speaker.
Recommendation: Accept in its present form after minor linguistic corrections.
Author Response
In this rewiev article the authors summarized progress in the research of epigenetic memories in hematopoietic stem cells and progenitor cells. The layout of the paper is typical for review articles. The choice of content does not raise objections. Citations are adequate to the content and clear. The 107 references are mostly from recent years. Due to some phrases it requires re-reading and correction by an English native speaker.
Recommendation: Accept in its present form after minor linguistic corrections.
Re: Thank you for your suggestion. We checked and modified phrases and sentences by ourselves, then the manuscript was checked by a native speaker using an English editing service.
Reviewer 3 Report
The manuscript by Aoyama et al excellently summarizes our current understanding of epigenetic memories in hematopoietic stem and progenitor cells. The manuscript appears to cover the subject quite comprehensively and adequately. I suggest a couple of points to improve the current manuscript.
1. I suggest the authors to revise their sentences carefully. I find numerous grammatical mistakes and awkward expressions in English. For example, “at a single resolution” should read “at the single resolution” in lines 25 and 26. “provides us genome-wide…” in line 46 should read “provides us with genome-wide…”. “Recently studies for gene regulation…” in line 59 should read “Recently, studies on gene regulation…”There are numerous other sentences that show grammatical and style errors. Correcting these would enhance readers’ comprehension of the manuscript.
2. A dedicated figure depicting ATAC-seq, Hi-C, and bulk and scRNA-seq are needed. The manuscript emphasizes the recent methodological development in NGS. Therefore, having a dedicated figure that introduces these methods will greatly help readers.
3. Figure legends are overall too short. I suggest the authors to expand the legends so that each figure legends can be self-explanatory. Also, adding more figure citations in the text should help readers follow the manuscript. Moreover, adding panel names (e.g., Figure 2A, 2B, etc) and citing each of these in the text may improve readers’ understanding of the figures and the text.
Author Response
The manuscript by Aoyama et al excellently summarizes our current understanding of epigenetic memories in hematopoietic stem and progenitor cells. The manuscript appears to cover the subject quite comprehensively and adequately. I suggest a couple of points to improve the current manuscript.
I suggest the authors to revise their sentences carefully. I find numerous grammatical mistakes and awkward expressions in English. For example, “at a single resolution” should read “at the single resolution” in lines 25 and 26. “provides us genome-wide…” in line 46 should read “provides us with genome-wide…”. “Recently studies for gene regulation…” in line 59 should read “Recently, studies on gene regulation…”There are numerous other sentences that show grammatical and style errors. Correcting these would enhance readers’ comprehension of the manuscript.
Re: Thank you for your suggestion. We checked and modified phrases and sentences by ourselves, then the manuscript was checked by a native speaker using an English editing service.
Reviewer 4 Report
Aoyama et al. present an interesting literature review on the study of epigenetic memory in hematopoietic stem and progenitor cells, focusing on the application of the most recent next-generation sequencing technologies. The issue is well introduced and arguments are appropriately presented. Manuscript is well written and exhaustively referenced and results suitable for publication.
Author Response
Aoyama et al. present an interesting literature review on the study of epigenetic memory in hematopoietic stem and progenitor cells, focusing on the application of the most recent next-generation sequencing technologies. The issue is well introduced and arguments are appropriately presented. Manuscript is well written and exhaustively referenced and results suitable for publication.
Re: We are grateful for the reviewer’s favorable comments.